# The Impact of Changing Climate on an Endangered Epiphytic Orchid (*Pleione formosana*) in a Montane Cloud Forest and the Conservation Challenge Ahead

**DOI:** 10.3390/plants13172414

**Published:** 2024-08-29

**Authors:** Rebecca C.-C. Hsu, Yi-Chiann Chen, Chienyu Lin

**Affiliations:** 1Forest Ecology Division, Taiwan Forestry Research Institute, Taipei 100051, Taiwan; cylin@tfri.gov.tw; 2Silviculture Division, Taiwan Forestry Research Institute, Taipei 100051, Taiwan; chiann@tfri.gov.tw

**Keywords:** ex situ conservation, in vitro germination, montane cloud forest, reintroduction biology, vascular epiphyte

## Abstract

*Pleione formosana* Hayata is an endemic orchid that was once widely distributed across the mid-elevations of Taiwan. However, populations of this orchid have steadily shrunk due to orchid poaching in most of its habitats. By correlating data from micrometeorological stations that we installed in the cloud forest canopy at the study site, Yuanyang Lake (YYL) from 2017, we discovered the critical role of spring rainfall in triggering flowering and summer rainfall in promoting the growth of new bulbs. We found that rising temperatures and frequent drought events threaten orchid growth, potentially leading to pathogen infections. We climbed old-growth yellow cypresses to collect seed capsules of *P. formosana* for in vitro germination at YYL beginning in the autumn of 2018. Orchid plantlets were reintroduced to the study site in mid-August of 2022. However, the seedlings failed to survive the summer of 2023. This study is the first persistent monitoring of this rare orchid in the forest canopy of this old-growth cloud forest. Based on the result, we propose conservation strategies and directions for protecting this orchid on a regional scale. Our study highlights the mounting challenge to conservation efforts posed by global climate change.

## 1. Introduction

Montane cloud forests (MCFs) are recognized as one of the most valuable ecosystems threatened by climate change from many perspectives [1,2,3,4]. First, the seasonal and diurnal changes in the microclimates of MCFs are generally small due to their constant envelopment in daily clouds, which moderate temperature fluctuations and sustain high humidity [5,6]. Therefore, many species inhabiting MCFs are adapted to relatively stable microclimates and sensitive to climate changes [7]. Second, as the climate warms, the altitude of cloud formation rises [2], and species, especially plants, may have difficulty migrating to keep pace with these changes. The cloud belt occurring at a specific elevation also isolates the species that grow within it, hindering interpopulation connection and promoting endemic specification [8]. Therefore, many MCF species are relatively rare, with fragmented distributions, which increase their vulnerability in the face of climate change [9].

Astride the Tropic of Cancer in the southwest Pacific, more than half of Taiwan’s land is over 1000 m above sea level [10], where moist ocean winds induce cloud formation and nurture a vast expanse of montane cloud forests [7]. Our previous study revealed that the montane cloud forest ecosystem of NE Taiwan may be the most severely impacted under various climate change scenarios compared to other forest types, threatening the suitable habitats of many epiphytic plants of this ecosystem [11]. Our species distribution models (SDMs) for the year 2100 found that cypress forests and many associated epiphytic plants will lose their suitable habitats. The forests of Chilan were one of the areas found to be in need of monitoring in our previous study [11]. The area is located in the Central Mountain Range in northern Taiwan, ranging in altitude from 1650 to 2444 m. It is a typical montane cloud forest ecosystem comprising pristine old-growth forests of mainly yellow cypress (*Chamaecyparis obtusa* var. *formosana* (Hayata)) (Figure 1). The area, including Yuanyang Lake (YYL), has been declared a nature reserve and is a long-term ecosystem research (LTER) site. The epiphyte inventory shows that YYL has the most abundant and diverse epiphytic community in Chilan [12].

YYL is one of the most intact habitats of *Pleione formosana* Hayata, hosting abundant populations in northern Taiwan (Figure 2). The epiphytic orchid *P. formosana* is endemic to Taiwan, growing on peat moss-covered branches or rocks, distributed between 1200 and 2500 m elevation in montane cloud forests throughout the island. The species was discovered and named in 1911 [13]. The white flower variety was initially described as a new species in 1933, but this was later corrected [14]. Although commercial nurseries cultivate the orchid for ornamental use, the wild population continues to shrink due to orchid poaching [15]. Moreover, populations of *P. formosana* are predicted to diminish dramatically under the future effects of climate change [16,17]. In 2017, we established a study site in YYL, recording the microclimate and monitoring populations of *P. formosana.* We also collected seed pods of *P. formosana* from the forest canopy and tested these for asymbiotic seed germination to preserve the species ex situ. We also planned to experiment with reintroduction procedures in the event that the species could be restored to its native habitats.

## 2. Material and Methods

### 2.1. Study Area and Species

The study was conducted at Yuanyang Lake (YYL), where the climate is generally warm, with a mean annual temperature of 13.9 °C and annual rainfall of ca. 4000 mm, which varies erratically with the torrential rains brought by typhoons in summer [18]. The winter NE monsoon brings intense fog and rain. In addition to vertical precipitation, fog water deposition is significant at YYL. This cloud forest is immersed in daily afternoon fog, and the area has recorded over 300 foggy days per year. A previous study found that annual fog deposition at YYL from 2003 to 2004 accounted for 10% of the total atmospheric hydrological input [19]. Persistent high air humidity at YYL nurtures abundant epiphyte communities that make up a conspicuous portion of the forest canopy.

The genus *Pleione* comprises ca. 26 species, mainly found in the Himalayan regions, SW China, and Taiwan [20]. Populations of *P. formosana* in Taiwan vary in leaf size, flower color, and the shapes of the keels on the lip [21]. Taxonomically, *P. formosana* belongs to the *P. bulbocodioides* (Franch.) Rolfe complex. Recent research using molecular information has found that *P. formosana* has at least two genetically diverse populations in Taiwan and is distinct from the population in SW China [15]. Most nurseries breed *P. formosana* by bulb propagation [22], and there are few cases of asymbiotic seed germination [23]. This is because bulb propagation produces good-quality flowering individuals in a relatively short period despite asymbiotic seed germination producing far more genetically diverse individuals.

### 2.2. Canopy Microclimate Measurement

The study began in December 2017. A micrometeorological station with loggers and sensors (Decagon devices, ICT International, Pullman, WA, USA) was installed about 20 m above ground in the forest canopy. The station measures and records wind velocity and direction, rainfall, air temperature, relative humidity (RH), and light intensity (using a HOBO Pendant UA-002-64) every 10 min (Figure 3). A visibility meter (MiniOFS model, Sten Löfving Optical Sensors, Göteborg, Sweden) was placed at a nearby site (3 km away) to record fog events every 30 min during the study. The visibility sensor outputs visibility as a voltage, where visibility of 1 km is represented by 1 volt, measuring up to 4 km, represented by 4 volts. A backup temperature/RH data logger (HOBO U23-001A, Onset Computer Corp., Bourne, MA, USA) was also installed in the canopy and on the forest floor. A ground weather station administered by the Central Weather Administration (CWA) is located within 1.5 km of YYL. We compared our canopy weather station data with CWA records to evaluate accuracy.

### 2.3. In Vitro Germination and Reintroduction

We have climbed yellow cypress trees to collect seed capsules of *P. formosana* every autumn since 2018 and brought intact seed capsules back to the lab for asymbiotic seed germination (Table 1). In the first three seasons, we germinated seeds at an orchid nursery in Yilan near the study site. Beginning in 2021, the capsules collected were brought back to the lab of the Taiwan Forestry Research Institute (TFRI) for asymbiotic seed germination (Figure 4). Capsules were surface-sterilized using 70% (*v*/*v*) ethanol for 1 min and 1% (*v*/*v*) sodium hypochlorite solution (NaOCl) with Tween 20 for 5 min under ultrasonication. Finally, the capsules were rinsed with autoclaved distilled water three times in a laminar flow cabinet, and moisture on the capsule surface was absorbed using sterilized filter paper. The capsules were cut longitudinally, and the seeds were equally distributed in bottles containing sowing medium (1/4 strength Murashige and Skoog (MS) medium [24], with 20 g L^−1^ sucrose, 2 g L^−1^ active charcoal, and 8 g L^−1^ agar, adjusted to pH 5.6 prior to autoclaving for 15 min at 121 °C). Seed-derived protocorms were transferred to seedling multiplication medium (1/2 MS, supplemented with 20 g L^−1^ sucrose, 8 g L^−1^ agar, 2 g L^−1^ activated charcoal, and 3 g L^−1^ Hyponex No. 1, 1 g L^−1^ tryptone, adjusted to pH 5.6 prior to autoclaving) to promote seedling growth and development [25]. Bottle seedlings were cultured in a growth chamber with the temperature controlled at 21 (±2) °C under a 12 h dark and 12 h light cycle at TFRI. In the summer of 2022, aggregate seedlings with 3 cm-long leaves and mini bulbs were reintroduced to natural peat moss at YYL.

### 2.4. Data Analysis

We calculated vapor pressure deficit (VPD) using a simple, accurate formula proposed by Huang [25], described in our previous study [12]. The formula takes into account both temperature and relative humidity (RH). Statistical analyses were conducted using R version 4.1.0 [26]. The mean climatic factors among different years were compared using an ANOVA test, and the correlation between visibility and temperature was assessed using Pearson correlation preceded by a Shapiro–Wilk normality test.

## 3. Results

### 3.1. Climate Recorded on Site

#### 3.1.1. Temperature and Rainfall from 2018

Summaries of rainfall and mean temperature during the spring (March and April) and summer (July and August) at YYL recorded by the CWA local meteorological station from 2018 are shown in Figure 5 and Figure 6. Spring rainfall ranged from 155 to 355 mm, with an average of ca. 250 mm. Rainfall for 2023 was only half of that of 2022 due to abnormal drought. Since 2018, there have been fewer typhoons across Taiwan, especially across NE Taiwan, which historically is most frequently affected by typhoons, so summer rainfall was significantly reduced at YYL over the last six years. In October of 2022, torrential rain brought by Tropical Storm Nesat, along with the NE monsoon, caused serious damage to the roads around YYL, so we stopped visiting study sites until March of 2023. In August of 2023, foehn winds brought by Typhoon Khanun caused dry leaves for most *P. formosana* individuals in YYL. Although the overall summer precipitation remained relatively unchanged (Table 1), the three-day mean temperature (from 5 to 7 August 2023) rose to 20.8 °C, with a maximum of 26.8 °C recorded, and the mean relative humidity dropped to a very low 76%. These conditions are stressful for the survival of *P. formosana*. Additionally, we noted a trend of reduced low-cloud events at YYL consistent with observations at other MCFs [3]. Our visibility records also showed a significant positive correlation (r = 0.59, *p* < 0.001) between daily mean temperature and visibility (i.e., fewer cloudy days, Figure 7).

#### 3.1.2. Canopy Sensors

We calculated the 24 h VPD and temperature differences for 2018 and the summer (July and August) of 2022 (Figure 8). Temperature changes are presented in line charts, and compared with 2018, the average summer temperature rose about 1.5 °C by noon and dropped 0.35 °C before dawn in 2022. The VPD (bar chart) increased more through the day and night in 2022 than in 2018, with the greatest increase (0.23) at 10 AM.

### 3.2. Observations on P. formosana Phenology and Field Collection

Beginning in 2018, we observed six flowering seasons and collected five seasons of seed capsules (Table 1, Figure 9). Seed capsule maturation dates varied from early September to early November, and the flowering dates determined the dates of mature fruits. We found fluctuated flowering and fruit mature dates rather than delayed or advanced trends. The flowering date was decided by the spring mean temperature and rainfall. In 2018 and 2023, a cold spring caused postponed flowering, while *P. formosana* flowered about five days earlier in 2019 than in 2009. Extreme weather and a heavy snow event in February 2018 delayed the flowering of *P. formosana* until May (usually in late March), and abnormally high spring temperatures brought by El Niño in 2019 accelerated fruiting by one month to August (normally in late September), so we only collected one capsule in 2019. Insufficient rainfall in March 2020, 2021, and 2023 resulted in poor flowering conditions in those years. In general, over the last 6 years, only 2022 was a rewarding season for fruit collection. Flowering was delayed in the spring of 2023, probably due to a cold and dry spring (Figure 5), compounded by the previous dry summer. In 2023, no mature fruits were observed in the field, making it the only year that we did not collect any capsules.

### 3.3. Growth of Seedlings and Reintroduction

We collected six capsules on 23 September 2021, while two capsules dehisced before germination. The average length (±SE.) of capsules was 2.82 (0.53) cm, and the average width was 0.85 (0.12) cm. In comparison with terrestrial orchids, we found that epiphytic *P. formosana* was relatively easy to germinate asymbiotically. All capsules collected in June 2022 failed to germinate (Table 1), suggesting that mature seeds were necessary for the successful germination of *P. formosana.* There was an absence of seed dormancy of *P. formosana*, and within 50 to 80 days of inoculation, mature seeds began to germinate and started to show healthy growth (Figure 10). We speculated that the bottle seedlings of the first 3 years (2018 to 2020) died gradually in summer due to lowland heat. Therefore, we cultured embryos in a growth chamber with the temperature controlled at 21 °C at the TFRI in 2021 and 2022, and the young grew quickly. On 17 August 2022, we introduced the plantlets germinated in 2021 to YYL (Figure 11). The young plants survived until 3 May 2023 but waned after the summer of 2023. The 2023 summer was particularly dry and hot due to extreme foehn winds brought by Typhoon Khanun.

## 4. Discussion

### 4.1. Key Climate Factors for P. formosana Survival

There is no constant change in direction from our six-season phenology observation, especially the flowering date of *P. formosana* (Table 1). In comparison with 2009, peak flowering dates shifted back and forward during the study period. Climatic records suggest that spring (March and April) rainfall is crucial for the flowering condition of the orchid. Summer rainfall helps the new bulbs grow, which is important for the next year’s flowering. Lower summer rainfall may have reduced the mature size of *P. formosana*’s bulbs, which we observed in the autumn of 2022. The following spring (2023) was the worst flowering season during the study period due to very low spring rainfall and small bulbs.

Table 1 shows the trend of rising maximum summer temperatures during the study period. Compared to 2018, the summer mean temperature increased about 1.5 °C at noon in 2022. There were 10 days in 2022 with daily maximum temperatures exceeding 25 °C, while there were no records above 25 °C in 2018. The extreme heat event was critically unfavorable to *P. formosana* survival; the ideal cultivation temperature at the orchid nursery is under 25 °C [27]. From our germination experiment, we also found that temperature is a decisive factor in the survival of *P. formosana*. Rising temperatures are especially harmful to seedling growth and will reduce the population of *P. formosana* in the future. We also noted an increased diurnal range of temperatures in 2022 (Figure 8). This is probably due to radiation cooling before dawn in the forest. Nights with clear skies result in steep temperature declines [7]. Although the R squared value of 0.35 is not particularly high, Figure 7 highlights the significant relationship between reduced clouds and rising air temperature. This suggests a more variable temperature regime of this cloud forest ecosystem in the future.

Our previous study found that the VPD is a decisive factor for epiphyte distribution in the studied forest [12]. The VPD takes into account both relative humidity (RH) and air temperature; therefore, it is an informative indicator of the key microclimate conditions for epiphytic plants [28]. Our results suggest stable VPD favors epiphyte diversity and abundance; however, there were also significantly rising mean and extreme values of VPD over the five years (Figure 12, ANOVA, *p* < 0.001). The mean VPD and standard deviations dramatically increased both in the forest canopy and on the forest floor, suggesting a trend of drought stress and unstable microclimates for *P. formosana* in the future.

### 4.2. Potential Threats from Changing Climate to P. formosana Survival

During the study period, we observed a decline in the population size of *P. formosana* in situ. From photos taken during the study period, we found many populations had disappeared from branches or trunks where they once grew. Since YYL is a conservation site, illegal poaching is unlikely to be the main reason for population decline. We also observed that populations of *P. formosana* near the forest floor sharply decreased. Two years ago, there were many individuals growing near the forest floor, but these have nearly all disappeared in recent years (field observation). Our data reveal that the VPD and mean temperature at the forest floor were lower than in the canopy, although both vertical forest strata exhibited an increasing trend (Figure 12, Appendix A Figure A1). If rising temperatures and the VPD were not the main reason for the disappearance of *P. formosana* near the forest floor, we can consider other influential microclimatic factors. Aside from light intensity, wind flow is one of the major differences between the forest canopy and floor. Raised garden beds are usually recommended for *P. formosana* nurseries [27] because elevated cultivation promotes air circulation and prevents pathogen infection in the orchid. Recent studies suggest that rising temperatures and high humidity may reduce disease resistance for native orchids [29]. A possible pathogen (Trichosphaeriaceae, unpublished data) that causes leaf spot in *P. formosana* was detected in the canopy soil at YYL [30]. However, strong airflow increases evapotranspiration in the forest canopy (Video, https://photos.app.goo.gl/dRFNvFVzsfjkHZQSA, accessed on 2 April 2024) and is probably unfavorable to the pathogen growth that infects *P. formosana*. We also found that individuals of the white flower variety have gradually disappeared at YYL. Further research is needed to determine whether the habitat niche of the white variety is more limited than that of pink flower individuals.

### 4.3. Implications for Re-Establishment of P. formosana

The *P. formosana* plantlets were reintroduced to YYL in mid-August of 2022. The onset of the NE monsoon (the first drop of air temperature below 10 °C) in 2022 was on September 8. The NE monsoon usually brings abundant rainfall to YYL, which enhances the survival of *P. formosana* seedlings. However, the reintroduced seedlings did not survive the following summer of 2023 and eventually shriveled. Snow events occur regularly at YYL, but the preliminary outcome of the reintroduction experiment suggests that summer heat rather than winter cold is probably the limiting factor for the survival of the species. The most suitable reintroduction season for YYL may be autumn rather than spring because the constant light rain brought by the NE monsoon provides the necessary moisture for young plants [30]. Moreover, to improve the survival rate, it is best to acclimatize the plantlets for a few months in the nursery on sphagnum moss after they are taken out from aseptic culture bottles [31].

## 5. Conclusions

This conservation project of *P. formosana* at YYL highlights the importance of the long-term monitoring of sensitive ecosystems in the face of anthropogenic climate change. Our six-year climate record reveals that climate change is not constant in direction, and it fluctuates greatly; therefore, it is difficult for forest organisms to adapt. Extreme weather events, such as drought or torrential rain, are becoming more likely with climate change and can have a greater impact on forest ecosystems than gradual global warming. The interaction between organisms of the forest ecosystem (e.g., microbial connections) may change with rising temperatures [32]. Preserving rare species in the face of global climate change is a mounting challenge to modern conservationists. An intact old-growth forest, such as the one at YYL, provides a micro-site with relatively stable habitats for sensitive species in a changing climate. Our report presents practical conservation measures to preserve the population of *P. formosana*, and our goal going forward is to focus on the management of local healthy populations of *P. formosana*.

## Figures and Tables

**Figure 1 plants-13-02414-f001:**
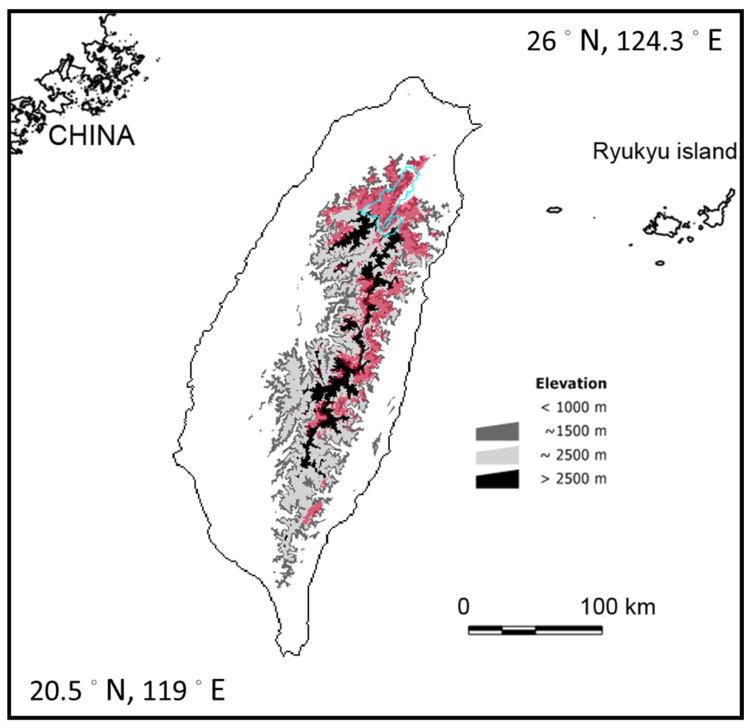
Distribution of cypress forests (red areas, MCFs) in Taiwan. Chilan, located in NE Taiwan (cyan boundary), comprises a vast area of cypress forests.

**Figure 2 plants-13-02414-f002:**
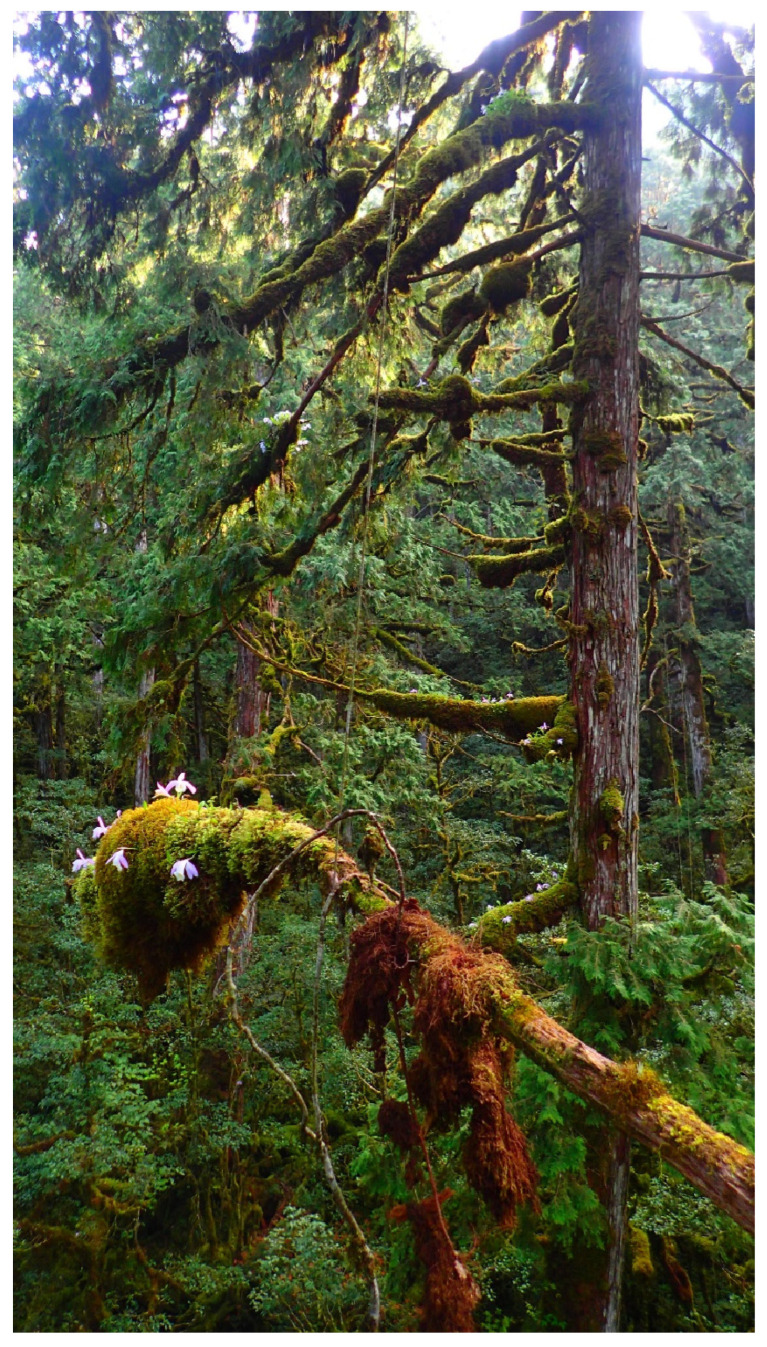
*P. formosana* in the canopy of old-growth yellow cypress forest at YYL.

**Figure 3 plants-13-02414-f003:**
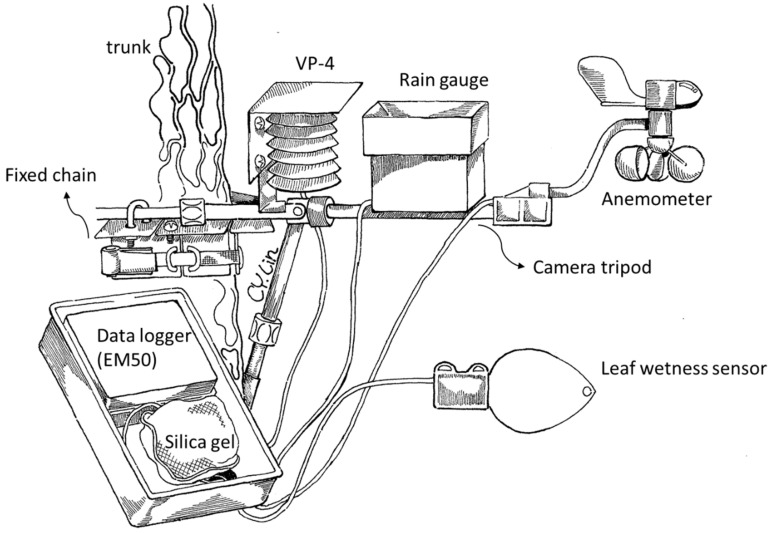
A canopy meteorological station was used in this study. An anemometer was fixed on a camera tripod facing north, with other sensors attached to the shared platform. All lines connected to sensors were routed to the bottom of an enclosed box and attached to the data logger inside. A pack of silica gel was left inside the box to keep the logger functioning reliably. All devices were attached to an adjustable steel platform mounted on a tree trunk.

**Figure 4 plants-13-02414-f004:**
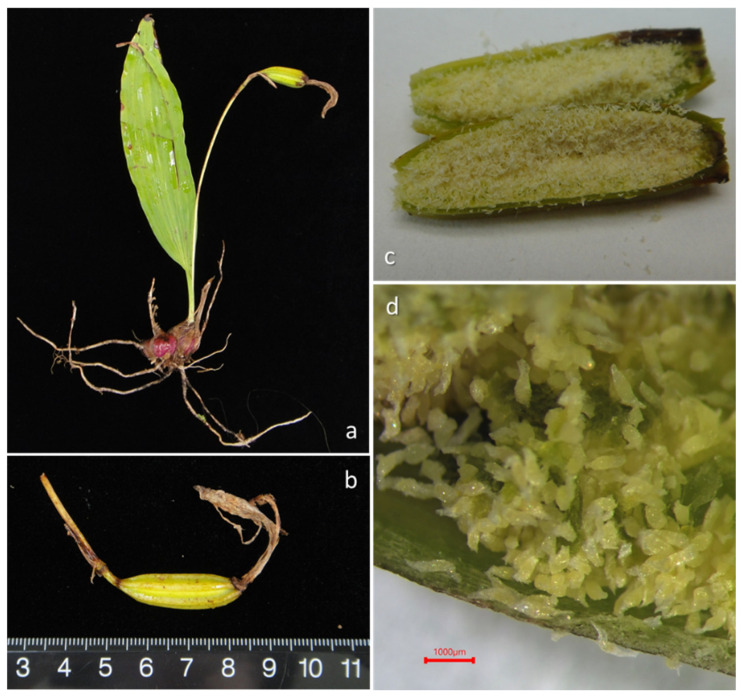
The adult *P. formosana* with a seed capsule and a new bulb formation (**a**), fully matured seed capsules were collected in late September before dehiscing (**b**), capsules were cut longitudinally after surface sterilized (**c**), and a healthy capsule contains many dusty seeds (**d**).

**Figure 5 plants-13-02414-f005:**
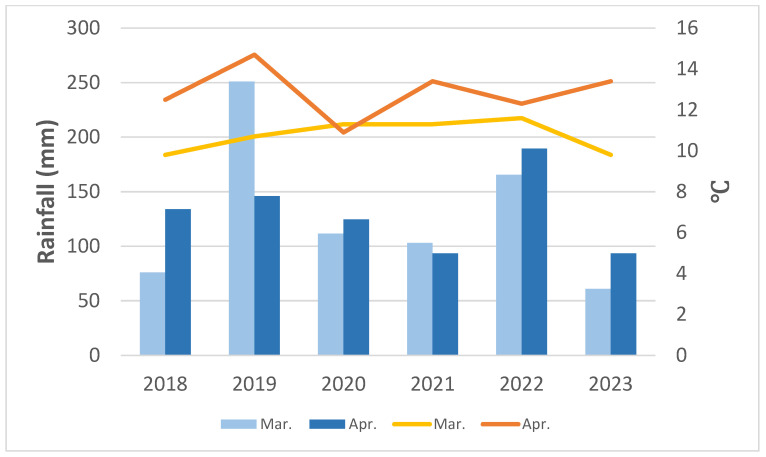
Spring rainfall (bars) and average temperature (lines) in YYL from 2018 to 2023 (CWA records).

**Figure 6 plants-13-02414-f006:**
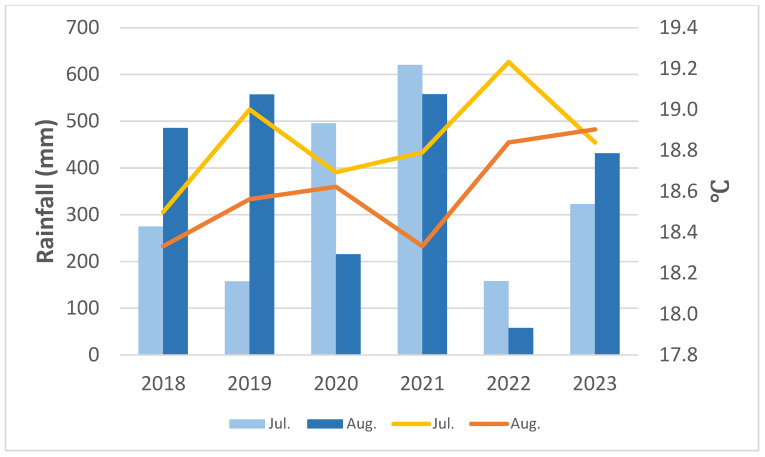
Summer rainfall (bars) and average temperature (lines) in YYL from 2018 to 2023 (CWA records).

**Figure 7 plants-13-02414-f007:**
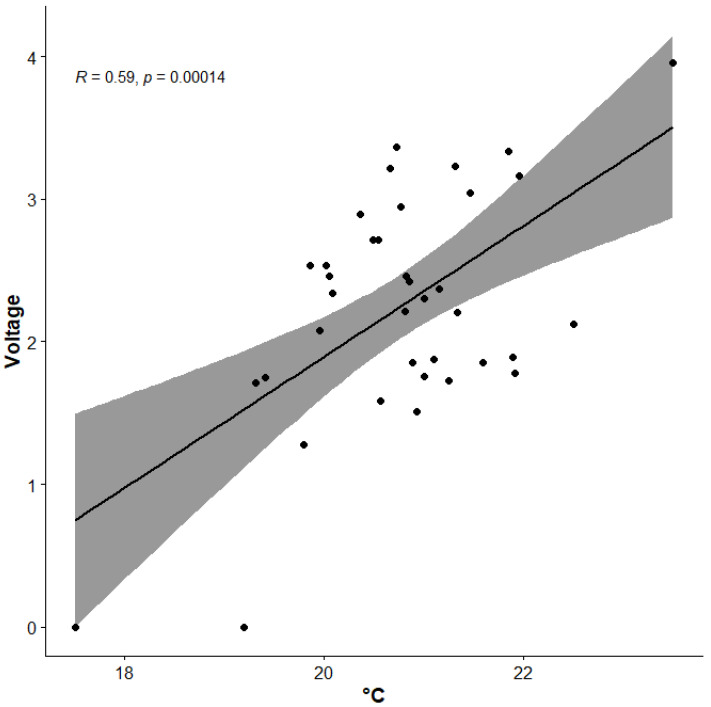
Pearson correlation between daily mean temperature (°C) and visibility (voltage) in 2020 summer. Data excluded rainy days with relative humidity (RH) over 95 percent.

**Figure 8 plants-13-02414-f008:**
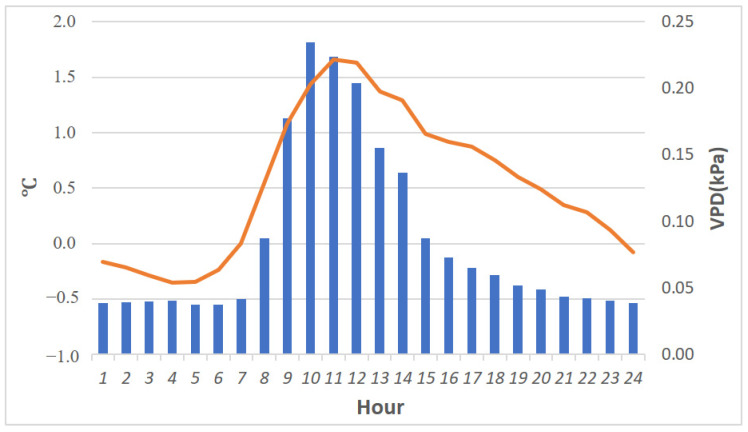
VPD (bar chart, kPa) and 24 h temperature (line chart, °C) difference between 2018 and 2022 summers (July and August).

**Figure 9 plants-13-02414-f009:**
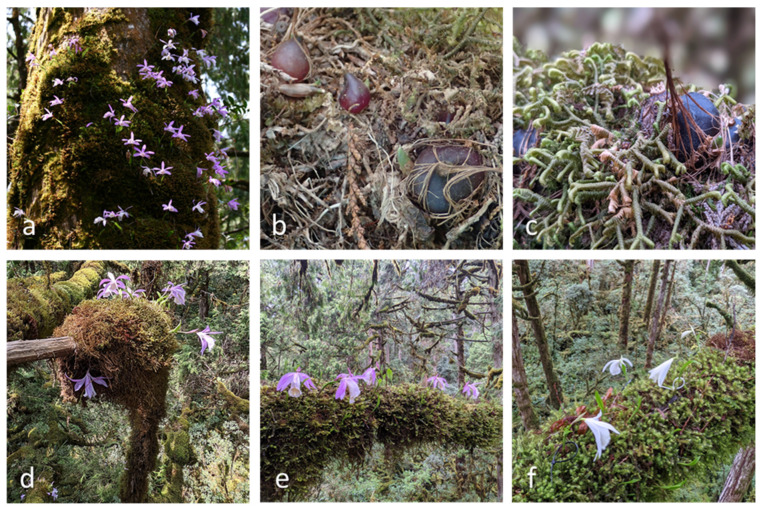
Delayed flowering was observed on 22 March 2018 (**b**) and 23 March 2023 (**c**). Compared with 31 March 2009 (**a**), flowering individuals have declined sharply over the last five years, as observed on 31 March 2021 (**d**) and 26 March 2022 (**e**), especially for the white variety, as observed on 26 March 2022 (**f**). Photo credit: Rebecca Hsu.

**Figure 10 plants-13-02414-f010:**
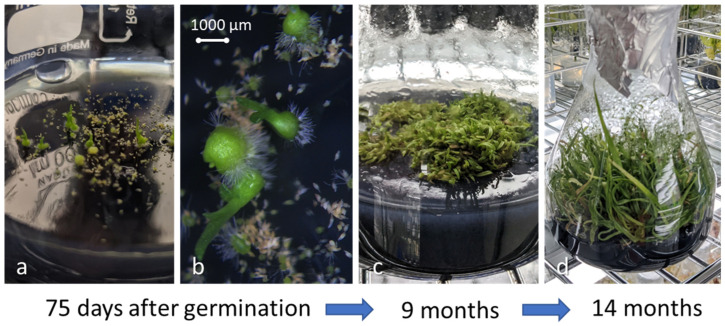
In vitro germination of *P. formosana* after 75 days (**a,b**), seedlings grew densely after nine months (**c**), and seedlings were ready to transplant after 14 months (**d**).

**Figure 11 plants-13-02414-f011:**
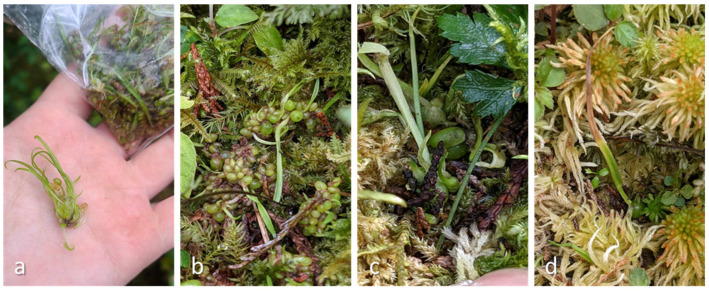
Plantlets were reintroduced to YYL in August of 2022 (**a**) but disappeared after the summer of 2023. September 2022 (**b**), March 2023 (**c**), May 2023 (**d**).

**Figure 12 plants-13-02414-f012:**
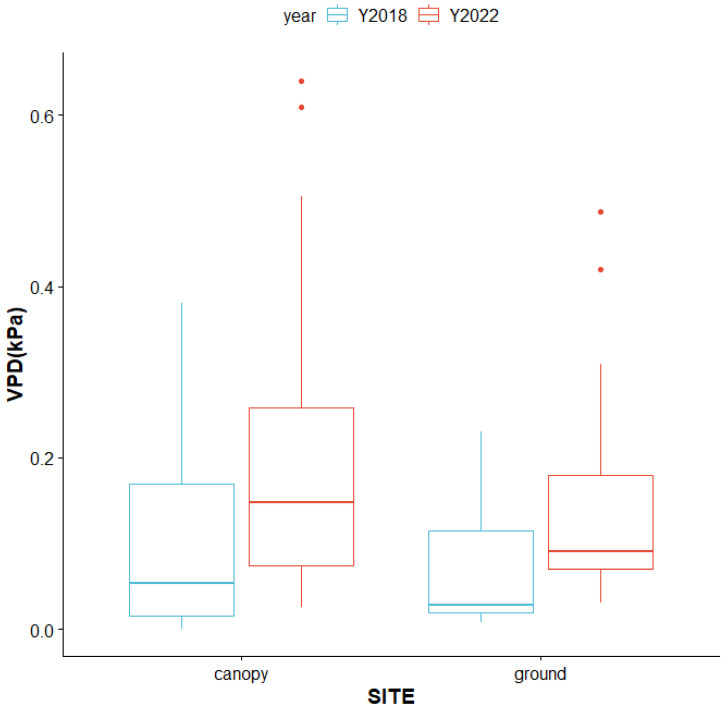
Boxplots (range with median line in the box) of mean VPD in 2018 (blue) and 2022 (red) in the forest canopy and on the forest floor (differences: sites, *p* < 0.05; year, *p* < 0.001).

**Table 1 plants-13-02414-t001:** Phenology of *P. formosana* observed from 2018 to 2023 at YYL. Bold text indicates extreme values during the study period. Canopy temperature records in the summer of 2020 were lost due to mechanical problems.

**Year**	2018	2019	2020	2021	2022	2023
**Flowering Date** ^1^	+1.1	−0.5	0	0	0	+0.7
**Flowers**	Average	Average	Few	Few	Average	Few
**Fruiting Date** ^1^	+3.3	−2	0	0	0	+1.1
**Fruits**	Average	Average	Few	Few	Abv. average	Few
**Spring**						
rainfall ^2^	210	**397**	236	**197**	355	154
temperature ^3^	−1.7/21.2	1.6/**25.5**	−0.5/21.3	2.2/21.2	−0.8/21.4	**−1.8**/24.2
**Summer**						
rainfall ^2^	761	715	711	**1178**	**216**	755
temperature ^3^	11.1/24.6	13.1/25.4	-	12.3/25.6	12/26.5	12.9/**26.8**
**Extreme** **Weather Event**	Snow event				Drought	Foehn wind
**Seed Collection Date**	4 7 November	1 10 September	2 * 9 October	6 ** 23 September	6 *** 21 June and 22 September	0

* One capsule dehisced before germination, ** two capsules dehisced before germination, *** capsules collected in June did not germinate. ^1^ Days/year compared with 2009, ^2^ mm, ^3^ min/max °C.

## Data Availability

The data presented in this study are available on request from the corresponding author. The data are not publicly available due to regulatory restrictions of the funding organization.

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
