# Peer review of "The Impact of Changing Climate on an Endangered Epiphytic Orchid (Pleione formosana) in a Montane Cloud Forest and the Conservation Challenge Ahead"

_plants, 2024, doi:10.3390/plants13172414_

Round 1

Reviewer 1 Report

Comments and Suggestions for Authors

p. 3. "10% of total hydrological input" Say where the measurements came from.

p. 6 - influence of Fohn wind - add "in YYL, although overall precipitation was not much affected"

Table 1. Align pod collection.

Fig. 8. Delayed flowering - say when photographs were taken.

Bottom p. 10.  How do you know that the 2022 will in fact be typical of the future? Add appropriate caveat.

11. "Two years previous" - do you have data to support this statement?

"although vertical stratification for both..." What does this mean?

Bottom. "HOWEVER, strong airflow..."

12. Implications for reestablishment OF P. formosana.

I wondered if there might be problems with the re-establishment of aseptic seedlings...

Conclusions - line 4. Of course, extreme events seem to become more likely with global warming, gradual although the overall trend may be.

Comments on the Quality of English Language

This reads quite well and clearly - I have made a couple of suggestions.

Author Response

Dear reviewer,

We really appreciate your valuable comments. We edited Table 1 based on your comments to quantify flowering and fruiting dates of Pleione formosana by + or - days/ year, and added min. and max. rainfall and temperature of the study period. You can find other responses addressed in the manuscript.

Kind regards,

Rebecca

Reviewer 2 Report

Comments and Suggestions for Authors

Dear authors,

I find extremaly interesting your paper as it is submitted. It presents evidence relevant. I find also appropriate your conclusions.

I have some minor concerns.

Figure 10 presents a Pearson correlation, whis is in it poor with R square 0.29 and this needs to be adressed in your comments. That correlation is poor is not bad per se but it should be commented.

Table 1 requires to be improved. In particular trends in phenological phases (flowering and fruiting) should be quantified in terms of + or - days/year. You can have an example in Annette's Menzel (2000) Trens in phenological phases in Europe between 1951 and 1996.

If you could somewhat quantifiy in other rows (Summer and Spring) by adding total rainfall per year and for temperatures (min-mena-max) could increase the accuracY

Also Table 1: notice that lowest row (Pods) present irregular alignment of columns

Author Response

Dear reviewer,

We really appreciate your valueable comments. We clarified and rewrote some sentences based on your suggestions. You can find addressed responses marked in red texts in the manuscript.

Kind regards,

Rebecca Hsu

Reviewer 3 Report

Comments and Suggestions for Authors

The manuscript deals with the conservation of Pleione formosana, an epiphytic orchid. The topic is interesting and within the scope of the journal. Unfortunately, the presentation is not good enough. The Introduction includes parts that shoul be moved to Results, and also some parts inserted in the Discussion are Results. In addition, the manuscript concentrates especially on the climate of the zone, while very few information is given on the plant species, on germination experiments, and so on. Some elaboration on the germination results was expected, also with some Figures on this. Therefore, the manuscript needs at least major revision, with the inclusion of the results on the germination experiments and in nature reinforcement of the population, and the discussion of these results. Also the Conclusion needs improvements, with a view on the international importance of this study and the future developments. See also the file in attachment, with my suggestions on minor points.

Comments on the Quality of English Language

Author Response

Dear reviewer,

We really appreciate your valuable comments. Following your suggestions, we re-organized some paragraphs, and added more information to the study. You can find addressed responses marked in red texts in the manuscript. 

Kind regards,

Rebecca Hsu

Round 2

Reviewer 3 Report

Comments and Suggestions for Authors

The authors addressed some of the issue arisen during the first revision round. However, the manuscript still presents the main highlighted problems. In particular, a figure and/or a table.on the germination tests would be a great improvement. In addition, the results, discussion and conclusion paragraphes were insufficiently amended. 

Comments on the Quality of English Language

Only minor issues.

Author Response

Dear reviewer,

We submit the revision of “The Impact of Changing Climate on an Endangered Epiphytic Orchid (Pleione formosana) in a Montane Cloud Forest and the Conservation Challenge Ahead” based on your round 2 comments. According to the comments, we add figure 4, redraw figure 10, and append capsules collection dates to Table 1. We also strengthen contents of Methods and Results to provide more information of germination experiments. In the Discussion and Conclusion, we address the critical impact from irregular high temperature (> 25 ℃), which is devastating to seedling development of P. formosana. Our reintroduction experiment of this orchid is still ongoing. We appreciate your effort to help us improve the manuscript. You can find addressed changes marked in red texts in the manuscript. And our responses to your round 1 comments is attached as PDF file.

Kind regards,

Rebecca Hsu

Round 3

Reviewer 3 Report

Comments and Suggestions for Authors

The authors improved the manuscript according to the suggestions. In my opinion, now the manuscript deserves publication.

Please, check the date "2009" in the following sentence in praragraph 3.2.:

In 2018 and 2023, a cold spring caused postponed flowering, while P. formosana flowered about five days earlier in 2019 than in 2009.

Comments on the Quality of English Language

English ok. Only some minor editing required.